# Enhanced Thermal Conductivity and Dielectric Properties of h-BN/LDPE Composites

**DOI:** 10.3390/ma13214738

**Published:** 2020-10-23

**Authors:** Lijuan He, Junji Zeng, Yuewu Huang, Xiong Yang, Dawei Li, Yu Chen, Xiangyu Yang, Dongbo Wang, Yunxiao Zhang, Zhendong Fu

**Affiliations:** 1College of Science, Harbin University of Science and Technology, Harbin 150080, China; hlj6607@163.com (L.H.); zjj941122@163.com (J.Z.); deardawei4li@hrbust.edu.cn (D.L.); cy374345509@163.com (Y.C.); Xiangyu_Yang1995@163.com (X.Y.); 2Key Laboratory of Engineering Dielectric and Its Application, Ministry of Education, Harbin University of Science and Technology, Harbin 150080, China; 3State Key Laboratory of Electrical Insulation and Power Equipment, School of Electrical Engineering, Xi’an Jiaotong University, Xi’an 710049, China; 15104548439@163.com; 4Department of Optoelectronic Information Science, School of Materials Science and Engineering, Harbin Institute of Technology, Harbin 150001, China; 5Tianjin Jinhang Technical Physics Institute, No. 58 Zhong Huan Xi Road, Tianjin Airport Economic Zone, Tianjin 300308, China; zhangyx0923@126.com (Y.Z.); fzd199@126.com (Z.F.)

**Keywords:** boron nitride, low-density polyethylene, thermal conductivity, dielectric properties, space charge

## Abstract

Low-density polyethylene (LDPE), as an excellent dielectric insulating material, is widely used in electrical equipment insulation, whereas its low thermal conductivity limits its further development and application. Hexagonal boron nitride (h-BN) filler was introduced into LDPE to tailor the properties of LDPE to make it more suitable for high-voltage direct current (HVDC) cable insulation application. We employed melt blending to prepare h-BN/LDPE thermally conductive composite insulation materials with different contents. We focused on investigating the micromorphology and structure, thermal properties, and electrical properties of h-BN/LDPE composites, and explained the space charge characteristics. The scanning electron microscope (SEM) results indicate that the h-BN filler has good dispersibility in the LDPE at a low loading (less than 3 phr (3 g of micron h-BN particles filled in 100g of LDPE)), as well as no heterogeneous phase formation. The results of thermal conductivity analysis show that the introduction of h-BN filler can significantly improve the thermal conductivity of composites. The thermal conductivity of the composite samples with 10 phr h-BN particles is as high as 0.51 W/(m·K), which is 57% higher than that of pure LDPE. The electrical performance illustrates that h-BN filler doping can significantly inhibit space charge injection and reduce space charge accumulation in LDPE. The interface effect between h-BN and the substrate reduces the carrier mobility, thereby suppressing the injection of charges of the same polarity and increasing the direct-current (DC) breakdown strength. h-BN/LDPE composite doped with 3 phr h-BN particles has excellent space charge suppression effect and high DC breakdown strength, which is 14.3% higher than that of pure LDPE.

## 1. Introduction

At present, high-voltage direct current (HVDC) transmission has shown a large-capacity, low-loss development trend, and people have put forward higher requirements for transmission reliability. However, with the increased transportation capacity of HVDC transmission systems, not only the internal space charge accumulation of the cable insulation layer but also the heat accumulation phenomenon during transportation will limit the further development and application of the cable [1,2]. Previous studies have pointed out that the accumulation of space charge in the cable insulation distorts the local electric field, which leads to partial discharge and even electrical breakdown of the insulation materials. Simultaneously, cable equipment generates a great deal of heat during transportation. The standard polymer insulation composites have low thermal conductivity, which makes it difficult to dissipate heat promptly, and eventually causes the internal temperature of the insulation material to rise [3,4,5]. If the long-term operating temperature of the cable equipment is over high, it may lead to accelerated insulation aging of the cable, shortened service life, and even thermal breakdown to damage the cable [6,7]. Although polyethylene is widely used for electrical equipment insulation due to its excellent mechanical properties, dielectric properties, corrosion resistance, and low price, pure polyethylene has poor heat dissipation properties, limiting its application in cable equipment preparation.

Until now, scholars have done a great deal of work to study new high-voltage DC transmission cable composite materials with excellent comprehensive properties. In the process of researching thermally conductive composite materials, they usually prepare composites doped with a high content of inorganic thermally conductive fillers. These composites have excellent thermal conductivity and corrosion resistance, but their electrical insulation properties are often lower than pure polymers [8,9,10]. Nonetheless, Zhang et al. found that the preparation of boron nitride nanosheets/styrene-(ethylene-co-butylene)-styrene tri-block copolymer/polypropylene (BNNS/SEBS/PP) composite insulation materials through multilayer hot pressing could significantly improve the electrical insulation and thermal conductivity of composite materials [11]. Du et al. studied the influence of the thermal conductivity of polyethylene/hexagonal boron nitride (PE/h-BN) with different mass fractions on its arc resistance. Studies have shown that when the mass fraction of h-BN fillers increases, the thermal conductivity and arc resistance of composites will be significantly improved [12]. Zha et al. pointed out that surface-modified MgO can effectively inhibit the accumulation of space charges in polypropylene/polyolefin (PP/POE) nano-composites. However, the thermal conduction paths of most insulating composite materials are poorly formed [13]. Therefore, it is necessary to carry out a large number of experiments to study and analyze the inorganic filler at a low filling amount to achieve high thermal conductivity and excellent electrical insulation. Hexagonal boron nitride (h-BN) is a graphene-like layered structure with excellent thermal conductivity (thermal conductivity of about 600 W/(m·K)), electrical insulation (5.9 eV wide bandgap), and thermal stability of inorganic fillers, so it has received widespread attention [14,15]. Therefore, the doping of h-BN fillers into low-density polyethylene (LDPE) to prepare polyethylene composite materials with excellent insulation and thermal conductivity properties may have broad application prospects in electrical equipment insulation and cables. 

Herein, we use melt blending to prepare h-BN/LDPE composites with high thermal conductivity and excellent electrical insulation. The morphology, thermal properties, and electrical properties of the composite film were investigated in detail. The space charge characteristics, electrical insulation properties, and thermal conductivity characteristics of low-density polyethylene materials filled with micron-scale h-BN filler were discussed. The results show that the thermal conductivity of the content is much improved, further elucidating the mechanism of improving thermal conductivity and suppressing space charge.

## 2. Materials and Methods

### 2.1. Materials and Reagents

Hexagonal boron nitride and low-density polyethylene were used as test materials. h-BN powder (1~2 μm, purity > 99.5%, 2.29 g·cm^−3^) was supplied by Aladdin Industries Ltd., Shanghai, China; LDPE (LD200BW, 0.920 g·cm^−3^) was produced by the Beijing Yanshan branch of Sinopec, Beijing, China.

### 2.2. Sample Preparation

The h-BN/LDPE composites were prepared by melt blending. The LDPE and micron-sized h-BN particles were weighed and mixed uniformly, and then the mixture was put into a torque rheometer (Harbin Hapro electric technology Co., Ltd., Harbin, China.) for blending, at a blending temperature of 120 °C. The content of micron-sized h-BN particles in the composite was 1, 3, 5, and 10 phr. The blended composite material was pressed on a plate vulcanizing machine (Kunshan Creator Testing Instrument Co., Ltd., Kunshan, China.) to form sheet samples of different thicknesses for performance testing. The pressing process was continuously conducted at 5, 15, and 25 MPa for 10 min each, and the temperature was 120 °C. Finally, the h-BN/LDPE composites were obtained by cooling with circulating water and placed in a vacuum drying box (Beijing Technol Technology Co., Ltd., Beijing, China) at a temperature of 70 °C for short-circuit drying for 24 h. The aluminum electrode (Jinan Longhai Aluminum Industry Co. Ltd., Jinan, China) was vapor-deposited on a vacuum coating machine (Beijing Technol Technology Co., Ltd., Beijing, China) according to the requirements of the related experiments to follow.

### 2.3. Characterization

Empyrean-type X-ray diffraction (Panalytical, Amelo, Netherlands) was used for phase analysis of the composite samples, with Cu Kα radiation. The infrared transmission spectrum of the sample was tested on an Equinox-50 Fourier transform infrared spectrometer (FTIR spectrometer) produced by Bruker, Karlsruhe, Germany. The effect of the doped h-BN filler on the chemical structure of LDPE and the dispersion of h-BN in the matrix were investigated by scanning electron microscope (SEM, Royal Philips Electronics, Amsterdam, Netherlands).

The thermal conductivity of the composite samples was tested using a laser flash thermal conductivity analyzer (LFA 447, NETZSCH Instrument Co., Ltd., Shanghai, China). The temperature range was 25~300 °C, and the thermal diffusion coefficient range was 0.1~1000 mm^2^/s. The specimens were 10 mm in diameter and 0.95 mm in thickness, spraying a thin layer of graphite powder (Fangda carbon New Material Co., Ltd., Lanzhou, China) on both sides of the sample.

The dielectric properties of the LDPE and h-BN/LDPE films were measured at room temperature by broadband dielectric spectrum (Novolcontrol, Berlin, Germany) in the frequency range from 10 Hz to 10^6^ Hz. Before the experiment, the composite samples were placed between two gold-plated stainless-steel electrodes(Jiangsu Jinzixuan Metal Technology Co., Ltd., Wuxi, China), 180 μm in thickness. The space charge distribution of the h-BN/LDPE composites was examined by pulsed electroacoustics (PEA, Shanghai Jiao Tong University, Shanghai, China. ) at room temperature. The space charge distribution in the sample was measured after the sample was subjected to a polarization field with strength of 40 kV/mm for 30 min, and the space charge decay rate of the sample after a short circuit for 30 min was quantitatively analyzed. The sample to be tested was a circular specimen with a diameter of 60 mm and a thickness of 300 μm.

The DC breakdown test was performed with a dielectric strength tester of a cylindrical electrode (YDZ-560, Yingkou Special Transformer Co., Ltd., Yingkou, China). The sample was placed between two cylindrical electrodes (Jiangsu Jinzixuan Metal Technology Co., Ltd., Wuxi, China) and wholly immersed in dimethyl silicone oil (Jinan Huanzheng Chemical Co., Ltd., Jinan, China) to suppress the flashover discharge, and the voltage was linearly increased at a step-up rate of 1 kV/s. In LDPE and h-BN/LDPE composites, the number of samples to be tested in each group was 12. We used the Weibull distribution to evaluate the breakdown strength of h-BN/LDPE composites with different contents. The Weibull statistical distribution was determined according to the following formula:(1)$$F\left(Ε\right)=1-\exp\left[-{\left(E/E_{0}\right)}^{\beta }\right]$$
where Ε is the electric breakdown strength measured experimentally, $F\left(Ε\right)$ is the accumulation probability corresponding to the parameter E, and $E_{0}$ is the characteristic breakdown strength, which describes the breakdown strength at 63.2% cumulative breakdown probability. The β is the shape parameter, which indicates the dispersion of experimental data. The larger the *β* in the DC breakdown experiment is, the smaller dispersion of the experimental data of the samples will be.

## 3. Results

### 3.1. Structure and Characterization

Based on X-ray diffraction, both LDPE and its composites (Figure 1) have characteristic peaks with 2*θ* values of 21.3° and 23.6°, corresponding to the diffraction peaks of the (110) and (200) planes of LDPE crystal, respectively. With the introduction of h-BN fillers, two characteristic diffraction peaks appear at 2*θ* = 26.6° and 43.6°, which correspond to the diffraction peaks of the (002) and (100) planes of h-BN crystal, respectively. By comparing the characteristic peaks of the composite material before and after attaching the h-BN filler, it is found that the crystal planes of the diffraction peaks corresponding to the h-BN filler and LDPE in the composite are consistent with the crystal planes of the standard diffraction peaks. It illustrates that the addition of h-BN filler will not significantly change the crystalline structure of LDPE, and no heterogeneous materials are generated during the hot pressing process of the blending of h-BN filler and LDPE.

As visible from the FTIR spectra, pure LDPE has several distinct characteristic peaks, and the absorption peak in the wavenumber range of 2848~2951 cm^−1^ is the symmetric and asymmetric telescopic vibration peak of C-H in -CH_3_. The two firm absorption peaks at 1462 cm^−1^ and 723 cm^−1^ are respectively the C-H shear vibration peak and the swing vibration peak in the polyethylene matrix [16]. Apart from the characteristic peaks of LDPE, the two sharp absorption peaks at 1423 cm^−1^ and 792 cm^−1^ of h-BN/LDPE composites are the stretching absorption vibration peak and vibration absorption peak in B-N in h-BN particles, respectively [17]. To gain insight into the functions of doping the h-BN filler on the LDPE matrix, the infrared spectrum of each sample was measured. As shown in Figure 2, doping the h-BN filler to the LDPE matrix did not change the position of the characteristic peak in the original LDPE. The distinct peaks of the h-BN/LDPE composite result from the superposition of those of the LDPE matrix and h-BN filler.

Figure 3 shows an SEM cross-sectional view of the LDPE and the h-BN/LDPE composite material. Compared with LDPE, there is almost no evident agglomeration in the cross section of the composite material doped with less than 3 phr h-BN particles, as shown in Figure 3a–c. Notably, as can be seen in Figure 3c, h-BN fillers have excellent dispersibility in the LDPE polymer. However, with the increase of the content of the h-BN filler, the volume of the inorganic phase increases. The microstructure of the composite becomes increasingly complex, and agglomeration occurs. As shown in the circled parts in Figure 3d,e, the composite material appears to be agglomerated.

### 3.2. Thermal Conductivity Test

The composites doped with h-BN fillers obtain better thermal conductivity, as shown in Figure 4. When the h-BN filler content is 10 phr, the thermal conductivity of h-BN/LDPE composites is up to 0.51 W/(m·K)—57% higher than that of LDPE (0.32 W/(m·K)). Although the thermal conductivity of the composite material was noticeably improved with the increase of the h-BN filler content, it may cause the insulation performance of the composite material to deteriorate. Research has shown that the useful improvement of the thermal conductivity of composites is closely related to the formation of thermally conductive networks of highly thermally conductive inorganic particles inside the composites [18,19]. At the same time, the observation results shown in Figure 3d show that h-BN cannot build a complete thermal conduction network in the composite material, although h-BN is partially accumulated in the composite at a relatively high content.

### 3.3. Dielectric Properties

#### 3.3.1. Permittivity and Loss Tangent

Figure 5 shows the frequency-dependent dielectric constant and dielectric loss of LDPE and different h-BN/LDPE composites. As shown in Figure 5a, the dielectric constant of LDPE is about 2.3 at 10–10^6^ Hz at room temperature, while the dielectric constants of the composite materials are slightly increased. The dielectric constants of composites with 1 and 3 phr h-BN particles only increased to 2.33 and 2.4, respectively. This shows that the dielectric constant of h-BN/LDPE composites is stable with frequency and does not change the low dielectric constant of LDPE. Figure 5b shows the relationship between the dielectric loss and the frequency of the composite material. The dielectric loss of the composite material is hardly affected by the h-BN filler, and remains in the range of 0.0004 to 0.0018. It has excellent dielectric properties and meets the insulation performance requirements of DC cables.

#### 3.3.2. Space Charge Test

Figure 6 illustrates the distribution of the space charge inside samples of pure LDPE and its composites at 1800 s with a polarization electric field of 40 kV/mm. As shown clearly from Figure 6a, when the electric field is 40 kV/mm and applied for 6 s, the space charge is injected into pure LDPE from anode and cathode. There is apparent hetero-polar charge accumulation near the cathode, and homogeneity-polarity charge accumulation is observed in the anode. As the polarization time reaches 900 s, the space charge around the cathode accumulates and forms a charge packet, and the charge of the same polarity near the anode decreases. When the polarization time reaches 1800 s, the charge packet near the cathode does not increase significantly.

As shown in Figure 6b, when the electric field is 40 kV/mm and is applied for 6 s, in the composite doped with 1 phr h-BN particles, there is a small amount of hetero-polar charge accumulation near the cathode, and homogeneity-polarity charge accumulation is observed near the anode. As the polarization time increases, the hetero-polar space charge decreases near the cathode. Figure 6c shows the space charge distribution of the 3 phr h-BN/LDPE composite. When the polarization time is 6 s, small heterogeneous charge packets appear near the anode and cathode. As the polarization time increases, the hetero-polar charge packet decays near the two poles. This is attributed to the excellent dispersion of h-BN in the LDPE, as shown in the SEM images in Figure 3b,c. Additionally, the traps formed at the contact interface between h-BN and the matrix can efficiently capture the injected charges to create an independent electric field, thereby reducing the effective electric field [20]. As shown in Figure 6d,e, when the polarization time is 6 s, there is apparent hetero-polar charge accumulation near the cathode and homogeneity-polarity charge accumulation near the anode. As the polarization time increases, the hetero-polar space charge decreases near the cathode, and the homogeneity-polarity charge near the anode decreases. However, the space charge peak of the composite material with a high loading of bn is also smaller than that of pure LDPE.

Figure 7 reveals the space charge distribution of LDPE and h-BN/LDPE composites within a short circuit of 1800 s after a polarized electric field of 40 kV/mm. It is evident from Figure 7a that at the depolarization time of 6 s, there are obvious hetero-polar charge packets near the pure LDPE cathode. Additionally, the peak value of the different polarity charge packet near the cathode is 8.96 C/m^3^. When the depolarization time increases in 900 s, the charge packets near the cathode decay. This may be because the charge density near the cathode in pure LDPE is high, and the concentration gradient is large, so the space charge decays rapidly with the increase of depolarization time. When the depolarization time reaches 1800 s, the charges of opposite polarity near the cathode decay further. As displayed in Figure 7a–e, when the depolarization time is within 1800 s, only a small amount of hetero-polar charge appears near the electrode of the h-BN/LDPE composite material, of which the space charge accumulation is much less than that of pure LDPE. 

In order to analyze the space charge variation and space charge decay in each sample, the average charge density Q and the average charge density decay rate V [21] are utilized. The calculation formulas are as follows:(2)$$Q=\frac{1}{L}\int_{0}^{L}\left|\rho \left(x,t,E\right)\right|dx=\frac{1}{x_{1}-x_{0}}\int_{x_{0}}^{x_{1}}\left|\rho \left(x,t,E\right)\right|dx$$
(3)$$V=\frac{\Delta Q\left(t\right)}{\Delta t}=\frac{\left|Q\left(t_{2}\right)-Q\left(t_{1}\right)\right|}{t_{2}-t_{1}}$$
where L is the thickness of the sample, $x_{1}$ and $x_{0}$ are respectively the positions of the cathode and anode electrodes, t is the depolarization time, $\rho \left(x,t,E\right)$ is the space charge distribution of the sample at position *x* received at short-circuit time t after a period of pressurization.$\;\Delta Q\left(t\right)$ is the attenuation amount of space charge density in a short time Δt. $Q\left(t_{1}\right)$ and $Q\left(t_{2}\right)\;$ are the average space charge densities in the sample at time $t_{1}$ and $t_{2}$ respectively.

The average volume charge density and decay rate of the LDPE and h-BN/LDPE composites during depolarization after poling at 40 kV/mm are presented in Figure 8. It is worth noting that the average volume charge density inside the pure LDPE sample is the largest. After the addition of the h-BN filler, the average volume charge density of the space charge in the composite material decreases significantly. However, the average volume charge density of each sample has a different decay rate with time. In pure LDPE, the decay rate of average volume charge density tends to be slow and then fast, which is due to the large charge concentration gradient of pure LDPE. Although the decay rate of the average volume charge density in h-BN/LDPE composites is first rapid and then slow, it is much lower than that of pure LDPE. Moreover, regardless of the average volume charge density or the decay rate, it is significantly lower than those of LDPE. The average volume charge density in composites decays at a moderate rate when doping h-BN particles at 3 and 10 phr.

On the one hand, because the amount of space charges accumulated is lesser in the h-BN/LDPE composite and the concentration gradient of the charges is small, its release rate is slow. On the other hand, when inorganic particles are introduced into the polymer, a large number of traps are formed at the interface between the polymer and inorganic particles [22]. Under the high-polarization electric field, the charge injected by the electrode will be trapped by the trap formed at the interface between the inorganic particles and the matrix, forming an independent reverse electric field, which increases the charge injection barrier and inhibits the accumulation of space charge [20,23]. Therefore, the more traps introduced by the composite material with a high loading of h-BN, and the less space charge injected into the composite material, the lower its average charge density. As shown in Figure 8b, the average charge density decay rate of the composites with 3 phr h-BN particles is the smallest.

#### 3.3.3. DC Breakdown Strength Test

It can be clearly observed from Table 1 and Figure 9 that the DC breakdown strength of pure LDPE is 298.2 kV/mm. After filling h-BN in the LDPE matrix, the DC breakdown strength *E*_0_ of the composite material first increases, reaches a maximum value at 3 phr, and then decreases. The composite materials with 1 and 3 phr h-BN filling materials have breakdown field strengths of 313 and 340.7 kV/mm, respectively, which are 4.96% and 14.3% higher than those of pure LDPE. This is primarily due to the uniform dispersion of h-BN particles in the LDPE matrix at low filling of h-BN particles. The interface between the h-BN filler and the substrate interaction region will introduce the effect of deep traps, reduce the carrier concentration and mobility, and suppress the space charge accumulation under a DC electric field [24,25,26]. Previous studies have noted that an essential factor leading to insulation failure under DC electric field is space charge accumulation, and the introduction of h-BN filler can suppress this. However, with the further addition of h-BN fillers (content > 3 phr h-BN particles), the breakdown strength E_0_ of the composite deteriorates, which may be caused by defects introduced inside the sample. Simultaneously, when the h-BN filler content is high, the h-BN particles show relatively low dispersion and relatively large agglomeration in the LDPE matrix. Filler agglomeration may act as a defect, worsen the electric field distortion around the filler, cause the partial discharge of the sample, and eventually lead to a decrease in the breakdown field strength of the composite material [27]. In short, composites doped with 3 phr h-BN particles have better breakdown characteristics.

## 4. Conclusions

To sum up, we prepared h-BN/LDPE composite materials with high thermal conductivity and excellent electrical insulation properties by melt blending. SEM images illustrate that h-BN particles are uniformly dispersed in the matrix at a low load content of 3 phr, demonstrating low dielectric constant, excellent space charge suppression, and high DC breakdown strength. In addition, the breakdown field strength is 14.3% higher than that of pure LDPE. This work provides a reliable, practical, and cheap scheme for the insulation materials of HVDC cables. Therefore, doped by an appropriate amount of h-BN filler, the thermal conductivity and electrical properties of the composite can be effectively improved, which provides an effective method for the preparation of HVDC cable insulation materials.

## Figures and Tables

**Figure 1 materials-13-04738-f001:**
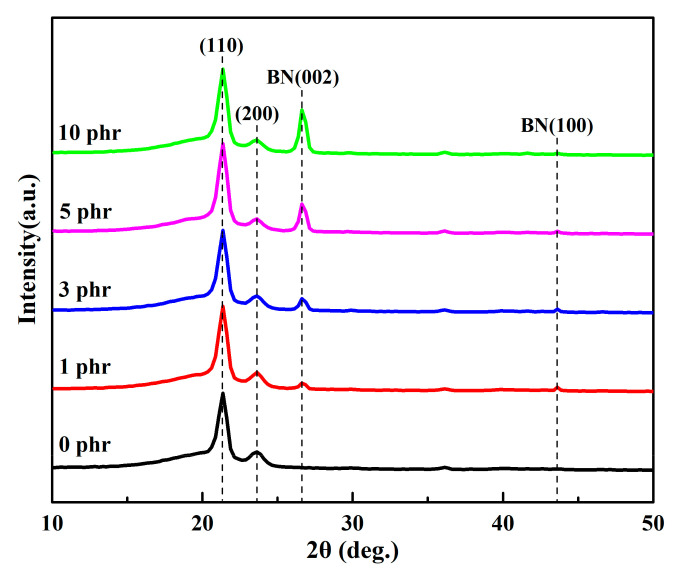
XRD images of h-BN/LDPE (hexagonal boron nitride/low-density polyethylene) composites with different h-BN fillers contents and LDPE.

**Figure 2 materials-13-04738-f002:**
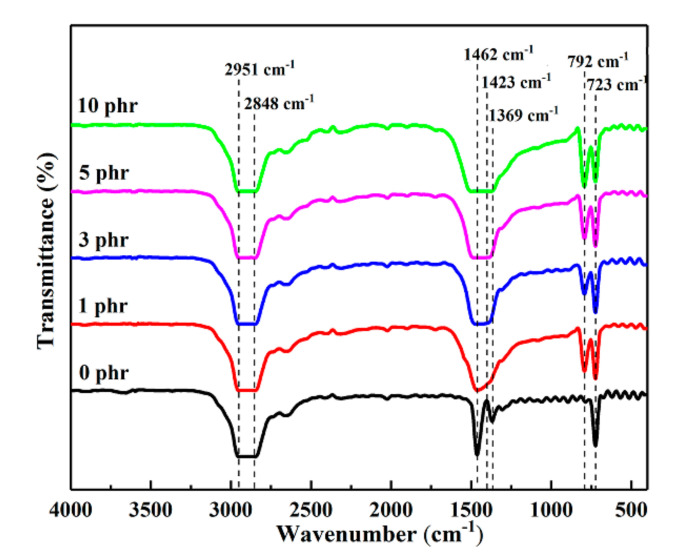
Fourier transform infrared spectra of LDPE and h-BN/LDPE composite materials.

**Figure 3 materials-13-04738-f003:**
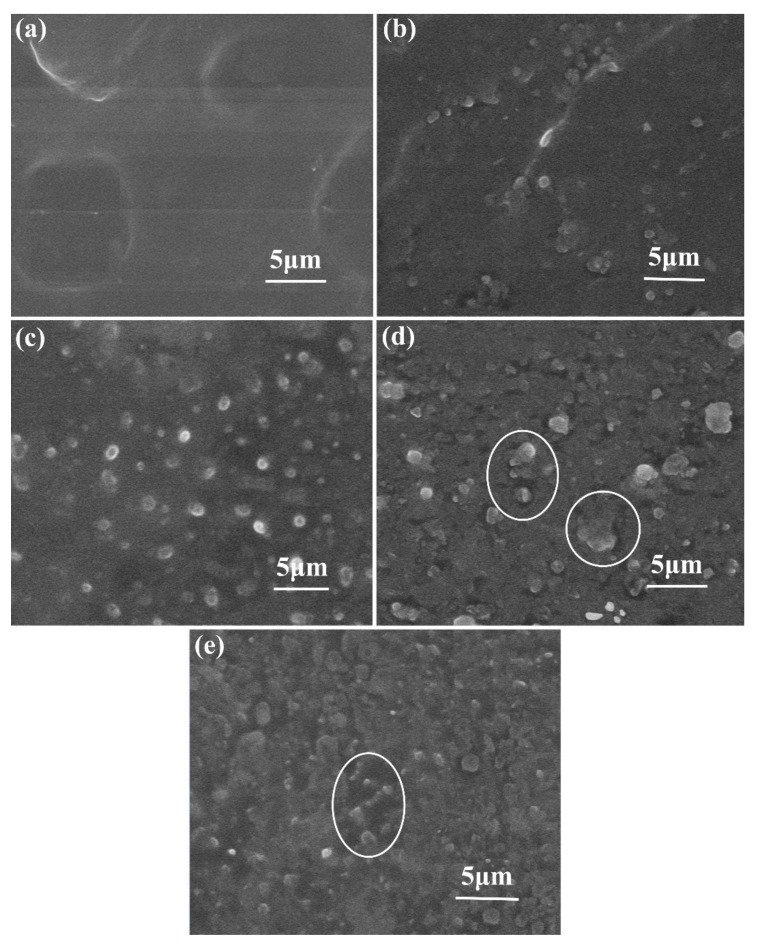
SEM cross-sectional view of h-BN/LDPE composite sample doped with h-BN fillers: (**a**) 0, (**b**) 1, (**c**) 3, (**d**) 5, (**e**) 10 phr.

**Figure 4 materials-13-04738-f004:**
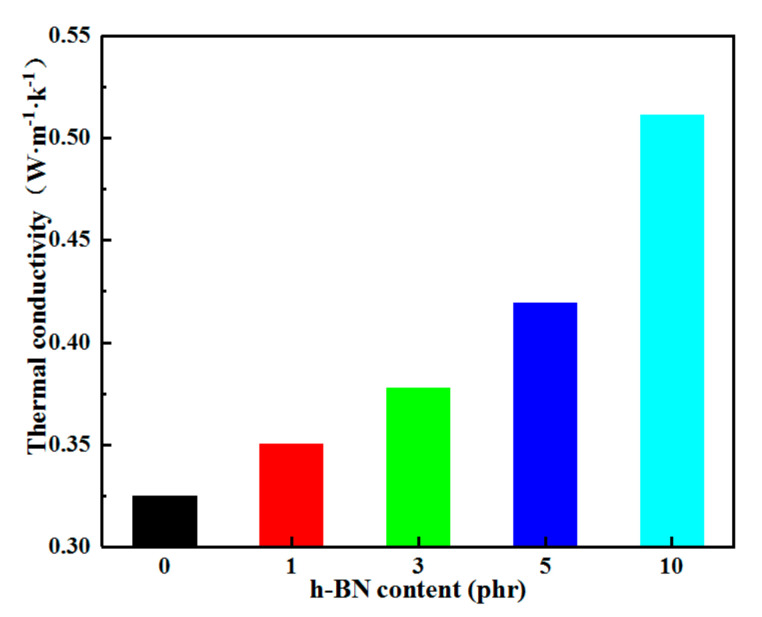
Thermal conductivity of LDPE and the composites.

**Figure 5 materials-13-04738-f005:**
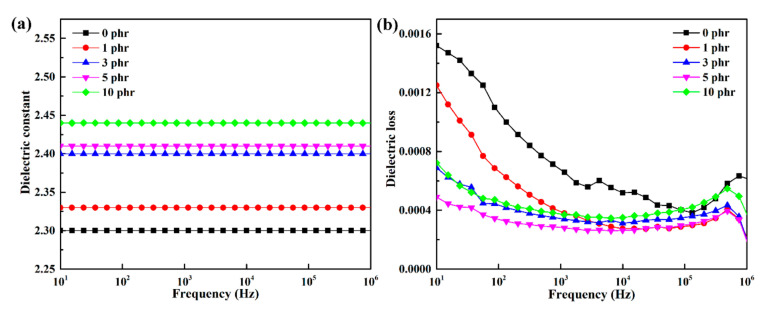
Relationship between (**a**) dielectric constant and (**b**) dielectric loss and frequency of h-BN/LDPE composites with different contents.

**Figure 6 materials-13-04738-f006:**
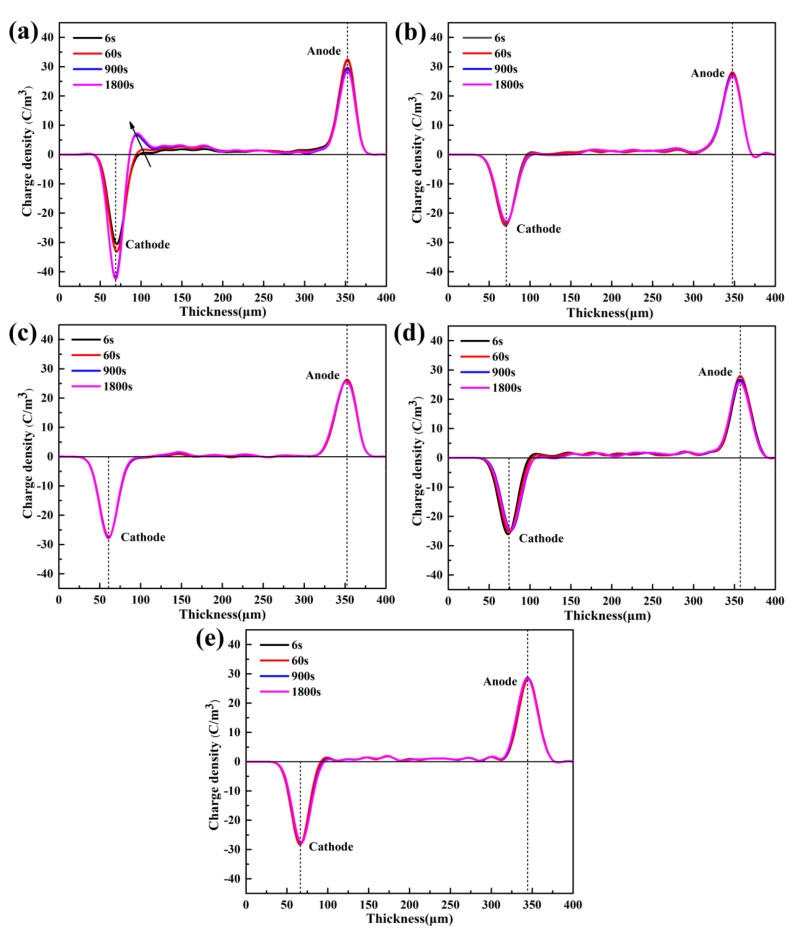
Space charge distributions of LDPE and h-BN/LDPE composites filled with h-BN: (**a**) 0, (**b**) 1, (**c**) 3, (**d**) 5, and (**e**) 10 phr under a polarization field of 40 kV/mm.

**Figure 7 materials-13-04738-f007:**
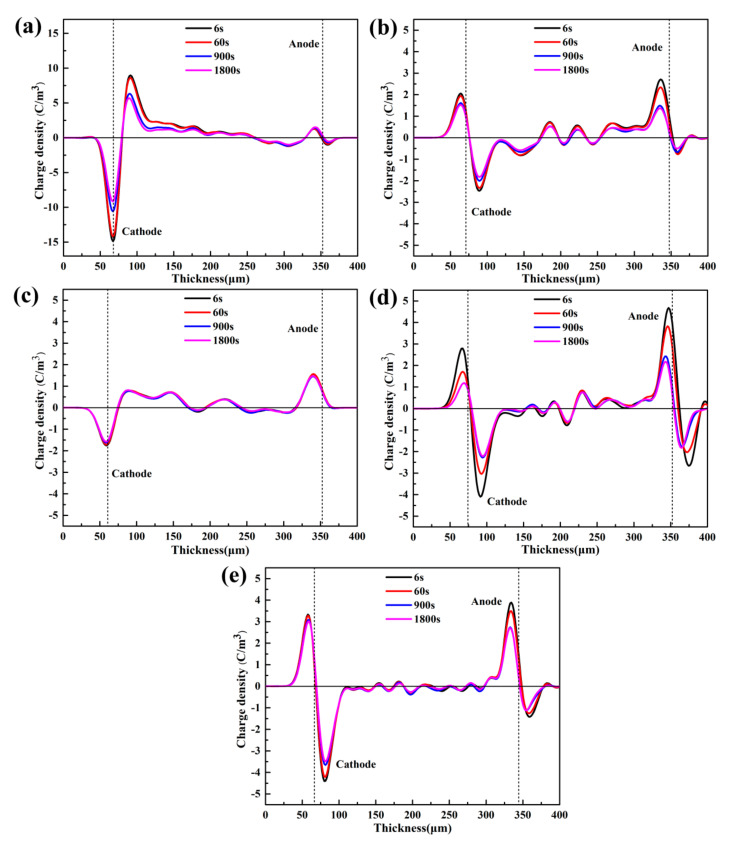
Space charge distribution of LDPE and h-BN/LDPE composites filled with h-BN: (**a**) 0, (**b**) 1, (**c**) 3, (**d**) 5, and (**e**) 10 phr during a short circuit.

**Figure 8 materials-13-04738-f008:**
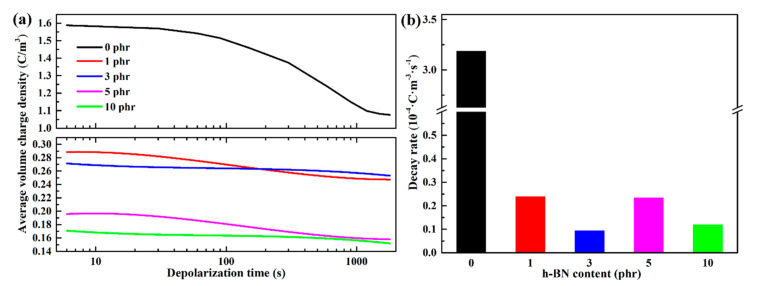
(**a**) The average volume charge density and (**b**) decay rate of the LDPE and h-BN/LDPE composites during depolarization after poling at 40 kV/mm.

**Figure 9 materials-13-04738-f009:**
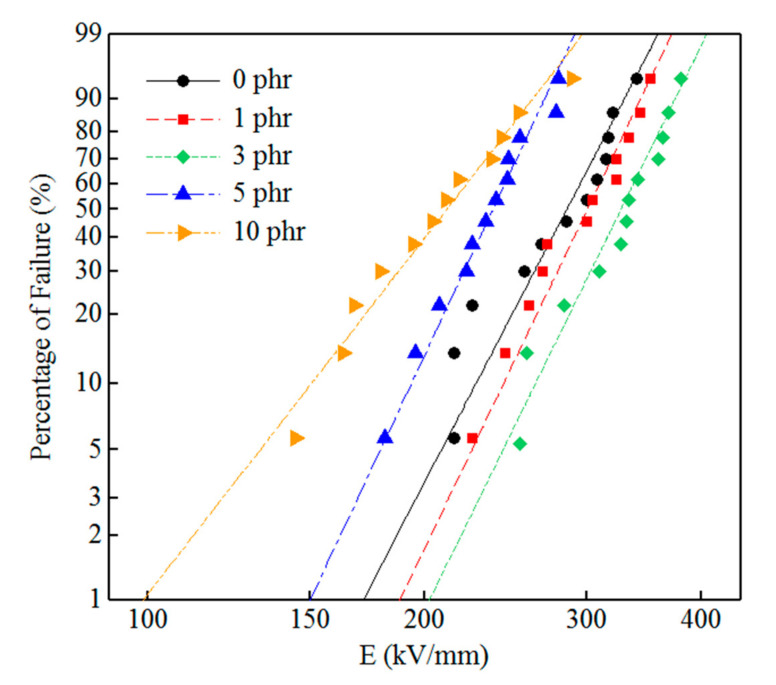
Weibull distribution of DC breakdown strength of LDPE and h-BN/LDPE composites with various h-BN filler contents.

**Table 1 materials-13-04738-t001:** Breakdown strength and Weibull distribution parameters of each sample.

Sample	Weibull Parameter	
	*β*	*E* _0_
LDPE	8.29	298.2
1 phr h-BN/LDPE	8.99	313.0
3 phr h-BN/LDPE	8.82	340.7
5 phr h-BN/LDPE	9.24	246.6
10 phr h-BN/LDPE	5.56	225.4

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
