# Peer review of "Enhanced Thermal Conductivity and Dielectric Properties of h-BN/LDPE Composites"

_materials, 2020, doi:10.3390/ma13214738_

Round 1

Reviewer 1 Report

Very interesting topic, including analysis of the internal structure and electrical properties concerning possible use for DC cables. In the future, it would be worthwhile to supplement the research with an analysis of mechanical properties. However, this is not a condition for the acceptance of this article.

Chapter 3.1, it would be useful to mention which methods correspond to the evaluation in the text. It is evident from the content, but it would be worthwhile (for example, line 133: Based on X-ray diffraction both LDPE ..., Line 145: As visible from FTIR spectra, pure LDPE ...).

Line 145: when speaking about peaks in the range 2848 – 2951 cm-1, it would be appropriate to measure with higher sensitivity. Here it seems to be a “totally absorbed” spectrum (peak maxima are “cut off”). Probably caused by the higher content of the sample compare to KBr powder during the preparation of the pills for transmission measurement or higher path length of the sample.

Figure 2: I assume that the y-axis is in units of transmittance (%Transmitance). It would be appropriate to add a description of the y-axis to the figure. Furthermore, it would be advisable to add a spectrum of the filler itself.  

Line 185: Only comment: Why the frequency 103 Hz is mentioned. Is it due to some relation to probably application or due to higher differences among measured data?

Author Response

Dear editor,

Re: Manuscript ID: materials-953093

Title: Enhanced Thermal Conductivity and Dielectric Properties of h-BN/LDPE Composites

Author(s): LiJuan He, JunJi Zeng, Yuewu Huang, Xiong Yang, Dawei Li, Yu Chen,

Xiangyu Yang, Dongbo Wang*, Yunxiao Zhang, Zhendong Fu.

Thanks for your favorite consideration and the reviewers’ insightful comments about my manuscript. Your comments and those of the reviewers were enabled us to greatly improve the quality of our manuscript. In the following pages are our point-by-point responses to each of the comments of the reviewers. Revisions in the revised manuscript are shown using red highlight for additions. Moreover, we have carefully checked the manuscript and corrected the English writing errors. We hope that the revisions in the manuscript and our accompanying responses will be sufficient to make our manuscript suitable for publication in Materials.

 Thank you very much for your time and consideration. I am looking forward to your response.

Yours sincerely,

Dongbo Wang

School of Materials Science and Engineering

Harbin Institute of Technology

Harbin 150001, China

E-mail: wangdongbo@hit.edu.cn

Reviewer #1 (Technical Comments to the Author):

Very interesting topic, including analysis of the internal structure and electrical properties concerning possible use for DC cables. In the future, it would be worthwhile to supplement the research with an analysis of mechanical properties. However, this is not a condition for the acceptance of this article.

  1. Chapter 3.1, it would be useful to mention which methods correspond to the evaluation in the text. It is evident from the content, but it would be worthwhile (for example, line 133: Based on X-ray diffraction both LDPE, Line 145: As visible from FTIR spectra, pure LDPE).

Reply: Thanks very much for the reviewers' suggestions. To make the expression more reasonable, we have revised the corresponding parts of the manuscript.

“Based on X-ray diffraction both LDPE and its composites (Figure 1) have characteristic peaks with the 2θ values of 21.3° and 23.6°” Detailed changes are marked out in red. (for details, see line 134-135 on page 3)

“As visible from FTIR spectra, pure LDPE has several distinct characteristic peaks...” Detailed changes are marked out in red. (for details, see line 146 on page 4)

“As shown in Figure 3, it is an SEM cross-sectional view of the LDPE” Detailed changes are marked out in red. (for details, see line 159 on page 4)

  1. Line 145: when speaking about peaks in the range 2848–2951 cm-1, it would be appropriate to measure with higher sensitivity. Here it seems to be a “totally absorbed” spectrum (peak maxima are “cut off”). Probably caused by the higher content of the sample compare to KBr powder during the preparation of the pills for transmission measurement or higher path length of the sample.

Reply: Thanks for the reviewers’ insightful comments about my manuscript. We agree with the reviewer’s point of view. peak maxima are “cut off”, It may be due to the higher content of the sample compared with KBr powder during the preparation of the pill for transmission measurement, or the longer path length of the sample.

  1. Figure 2: I assume that the y-axis is in units of transmittance (%Transmitance). It would be appropriate to add a description of the y-axis to the figure. Furthermore, it would be advisable to add a spectrum of the filler itself.

Reply: Thanks for the reviewers’ suggestions. To make the expression more accurate, we add a description of the y-axis to the figure in the manuscript.

This manuscript r is mainly to study the thermal conductivity and dielectric properties of the enhanced thermally conductive h-BN/LDPE composite. Also, it can be seen in the figure that the two sharp absorption peaks at 1423 cm-1 and 792 cm-1 of h-BN/LDPE composites are the stretching absorption vibration peak and vibration absorption peak in B-N in h-BN particles, respectively [R1].

Detailed changes are marked out in red. (for details, see Figure 2 on page 4)

Figure R2. Fourier transform infrared spectra of LDPE and h-BN/LDPE composite materials.

References

  • Kiho Kim, Myeongjin Kim, Jooheon Kim. Enhancement of the thermal and mechanical propertyes of a surface-modifed boron nitride-polyurethane composite. Polymers for Advanced Technologies. 2014, 25, 791–798.
  1. Line 185: Only comment: Why the frequency 103 Hz is mentioned. Is it due to some relation to probably application or due to higher differences among measured data?

Reply: Thanks for the reviewers' suggestions, we have modified this part in the manuscript. Besides, we have changed the revised expression of this sentence, which is as follows. As visible from Figure 3, the dielectric constant of LDPE is about 2.3 at 10-106 Hz at room temperature. Detailed changes are marked out in red. (for details, see line 187-188 on page 6)

Figure R2. Relationship dielectric constant and frequency of h-BN/LDPE composites with different contents.

Reviewer 2 Report

The presented work addresses important improvement achieved upon loading LDPE with hBN particles. The results have important application value, therefore I suggest to accept this work for publication after correcting the following comments:

From SEM images we note that there are agglomerations of hBN nanoparticles. This imply some fluctuations in material properties. However, no information is provided on the number of samples tested. Please inform the reader about the number of samples, which were tested for particular measurement. 

Fig.3 should present also pristine LDPE for comparison.

line 203: "Meanwhile, the injected space charge gradually migrates to the inside of  the material." - This statement requires an explanation of facts, which were used to conduct this conclusion

line 205: "To evaluate the performance of the h-BN filler, we introduced that into the LDPE matrix. When the composite material is doped with 1 phr h-BN particles, as shown in Figure 6b, it can effective suppress space charge accumulation of the composite." - To my opinion, these statements are general and belong to introduction. 

line 209: "Figure 6c shows that there is no visible space charge accumulation in the composites with 3 phr h-BN particles near the two poles." - The statement seems to be incorrect - Fig. 6b is similar to 6c.

Figures 6 and 7: Please add description of physical quantity in the legend. 

line 258: "As the charge undergoes trapping, de-trapping, and re-trapping during the transfer process, deep traps make it more difficult for the traps to be trapped." - please note that traps can be filled or empty, but not trapped

line 211: "the traps formed at the contact interface between h-BN and the matrix can efficiently capture the injected charges to create an independent electric field, thereby reducing the effective electric field." - This hypothesis is of crucial importance in explanation of reduction of breakdown voltage. Also line 258 addresses traps in the same context. However authors doesn't explain, why, if the traps are formed at the interface between LDPE and hBN, the lowest charge density is observed in higher loading of hBN as presented in Fig.8.

Please check text for typos.

Author Response

Reviewer #2 (Remarks to the Author):

The presented work addresses important improvement achieved upon loading LDPE with hBN particles. The results have important application value, therefore I suggest to accept this work for publication after correcting the following comments:

  1. From SEM images we note that there are agglomerations of hBN nanoparticles. This imply some fluctuations in material properties. However, no information is provided on the number of samples tested. Please inform the reader about the number of samples, which were tested for particular measurement.

Reply: Thanks for the reviewers' suggestions. According to the reviewer’s opinion, we have added the number of samples tested in the DC breakdown part. In LDPE and BN/LDPE composites, the number of samples tested in each group is 12. Detailed changes are marked out in red. (for details, see line 122-123 on page 3)

  1. 3 should present also pristine LDPE for comparison.

Reply: Thanks for the reviewers' suggestions. To make the date more persuasive, the microstructure of pure LDPE was retested. The cross-sectional SEM images of LDPE and h-BN/LDPE composites are shown in Figure 1. The cross-sectional SEM images of LDPE have been added in our revised manuscript (for detail, see Figure 3 on page 5). Besides, the corresponding discussions have been added to the manuscript. Detailed changes are marked out in red. (for details, see line 158-160 on page 4).

Figure R1. The SEM images of LDPE synthesized in our manuscript

  1. line 203: "Meanwhile, the injected space charge gradually migrates to the inside of the material." This statement requires an explanation of facts, which were used to conduct this conclusion.

Reply: According to the reviewer’s suggestion, we further explained that the internal space charge of pure LDPE changes with the increase of polarization time. As shown clearly from Figure R1 that when the electric field is 40 kV/mm and applied for 6 s, the space charge is injected into pure LDPE from anode and cathode. There is apparent hetero-polar charge accumulation near the cathode, and the homogeneity-polarity charge accumulation is observed in the anode. As the polarization time reaches 900 s, the space charge around the cathode accumulates and forms a charge packet, and the charge of the same polarity near the anode decreases. When the polarization time reaches 1800 s, the charge packet near the cathode does not increase significantly. Detailed changes are marked out in red. (for details, see line 203-207 on page 6)

Figure R1 Space charge distribution of LDPE in the 40kV/mm field.

  1. line 205: "To evaluate the performance of the h-BN filler, we introduced that into the LDPE matrix. When the composite material is doped with 1 phr h-BN particles, as shown in Figure 6b, it can effective suppress space charge accumulation of the composite." To my opinion, these statements are general and belong to introduction.

Reply: Thanks very much for the reviewer’s suggestion. According to the reviewer’s opinion, we have modified this section. Detailed changes are marked out in red. (for details, see line 207 on page 6)

  1. line 209: "Figure 6c shows that there is no visible space charge accumulation in the composites with 3 phr h-BN particles near the two poles." - The statement seems to be incorrect - Fig. 6b is similar to 6c.

Reply: According to the reviewer’s suggestion, to make the expression in the manuscript more accurate, we have revised the corresponding parts in the manuscript. As shown in Figure 6b, when the electric field is 40 kV/mm and applied for 6 s, in the composite doped with 1 phr h-BN particles, there is a small amount of hetero-polar charge accumulation near the cathode, and the homogeneity-polarity charge accumulation is observed near the anode. As the polarization time increases, the hetero-polar space charge was decreases near the cathode. It can be observed that the injected charge from the cathode moves into the sample as the polarization time increases. Figure 6c shows the space charge distribution of the 3 phr h-BN/LDPE composite. When the polarization time is 6s, small heterogeneous charge packets appear near the anode and cathode. As the polarization time increases, the hetero-polar charge packet was decaying near the two poles.

Detailed changes are marked out in red. (for details, see line 207-213 on page 6)

Figure R1. Space charge distributions of LDPE and h-BN/LDPE composites filled with h-BN: (a) 1 and (b) 3 phr under a polarization field of 40 kV/mm.

  1. Figures 6 and 7: Please add description of physical quantity in the legend.

Reply: Thanks for the reviewer’s suggestion. According to reviewer’s opinion, we have added a description of the physical quantities in Figures 6 and 7.

"Figure R1 reveals the space charge distribution of LDPE and h-BN/LDPE composites within a short circuit of 1800 s after a polarized electric field of 40 kV/mm. It was evident from Figure 7a that at the depolarization time of 6 s, there are obvious hetero-polar charge packets near the pure LDPE cathode. And the peak value of the different polarity charge packet near the cathode is 8.96 C/m3. When the depolarization time increases in 900 s, the charge packets near the cathode decay. It may be because the charge density near the cathode in pure LDPE is high, and the concentration gradient is large, so the space charge decays rapidly with the increase of depolarization time. When the depolarization time reaches 1800 s, the charges of opposite polarity near the cathode further decay. As displayed in Figures 7a-7e, when the depolarization time is within 1800 s, only a small amount of hetero-polar charge appears near the electrode of the h-BN/LDPE composite material, of which the space charge accumulation was much less than that of pure LDPE.

To analyze the space charge variation and space charge decay in each sample, the average charge density  and the average charge density decay rate  [20] are utilized"

Detailed changes are marked out in red. (for details, see line 207-221 and line 226-236 on page 7-8)

Figure R1. Space charge distribution of LDPE and h-BN/LDPE composites filled with h-BN: (a) 0, (b) 1, (c) 3, (d) 5, and (e) 10 phr during short circuit.

  1. line 258: "As the charge undergoes trapping, de-trapping, and re-trapping during the transfer process, deep traps make it more difficult for the traps to be trapped." please note that traps can be filled or empty, but not trapped.

Reply: According to the reviewer’s opinion, we have revised the corresponding parts of the manuscript. Detailed changes are marked out in red. (for details, see line 263-267 on page 10)

  1. line 211: "the traps formed at the contact interface between h-BN and the matrix can efficiently capture the injected charges to create an independent electric field, thereby reducing the effective electric field." This hypothesis is of crucial importance in explanation of reduction of breakdown voltage. Also line 258 addresses traps in the same context. However authors doesn't explain, why if the traps are formed at the interface between LDPE and h-BN, the lowest charge density is observed in higher loading of h-BN as presented in Fig.8.

Reply: Thanks very much for the reviewer’s suggestion.

When inorganic particles are introduced into the polymer, a large number of traps are formed at the interface between the polymer and inorganic particles [R1]. Under high polarization electric field, the charge injected by the electrode will be trapped by the trap formed at the interface between the inorganic particles and the matrix, forming an independent reverse electric field, which increases the charge injection barrier and inhibits the accumulation of space charge [R2,R3]. Therefore, the more traps introduced by the composite material with high loading of BN, and the less space charge injected into the composite material, the lower its average charge density.

Detailed changes are marked out in red. (for details, see line 263-269 on page10)

References

  • Le Roy S, Segur P, Teyssedre G, Laurent, C. Description of bipolar charge transport in polyethylene using a fluid model with a constant mobility: model prediction. Journal of Physics D: Applied Physics, 2004, 37, 298-
  • LI S T, MIN D M, WANG W W. Modelling of dielectric break-down through charge dynamics for polymer nanocomposites. IEEE Transactions on Dielectrics and Electrical Insulation, 2017, 23, 3476-
  • Huang X , Jiang P , Yin Y . Nanoparticle surface modification induced space charge suppression in linear low density polyethylene. Applied Physics Letters, 2009, 95,:1546.

Round 2

Reviewer 2 Report

Thanks for your reply.

This manuscript is a resubmission of an earlier submission. The following is a list of the peer review reports and author responses from that submission.